# Cross-Oxygen Gradients Transcriptomic Comparison Revealed the Central Role of MAPK and Hippo in Hypoxia-Mediated Mammary Proliferation Inhibition

**DOI:** 10.3390/antiox13030288

**Published:** 2024-02-26

**Authors:** Zhenzhen Hu, Yi Lu, Jie Cai, Jianxin Liu, Diming Wang

**Affiliations:** Institute of Dairy Science, MoE Key Laboratory of Molecular Animal Nutrition, College of Animal Sciences, Zhejiang University, Hangzhou 310058, China; 12117005@zju.edu.cn (Z.H.); 21917023@zju.edu.cn (Y.L.); zjcaijie@zju.edu.cn (J.C.); liujx@zju.edu.cn (J.L.)

**Keywords:** mammary epithelial cell, cell proliferation, hypoxia, oxidative stress, MAPK signaling pathway, Hippo signaling pathway

## Abstract

The role of hypoxia in terms of affecting mammary epithelial cells (MECs) proliferation is closely associated with the milk synthesis of lactating mammals. Primary bovine MECs were cultured at 1, 6, 11, 16, and 21% O_2_ for 24 h. The results showed that cell proliferation decreased linearly, and hypoxic inducible factor (HIF)-1α expression increased linearly along with the declining O_2_. The linear increase in oxidative stress resulted in the accumulation of malondialdehyde and reactive oxygen species and decreased antioxidant enzyme activities following the reduced O_2_. Concerning mitochondria, the dynamin-related protein 1 showed improved expression, and optin atrophy protein 1 decreased along with the decreasing O_2_ gradient, which led to decreased mitochondrial mass and mitophagy emerging under 1% O_2_. Oxygen concentration-trend RNA-seq analysis was conducted. Specifically, HIF-1-MAPK (1% O_2_), PI3K-Akt-MAPK (6% O_2_), and p53-Hippo (11 and 16% O_2_) were found to primarily regulate cell proliferation in response to hypoxia compared with normoxia (21%), respectively. In conclusion, our study suggests that bMEC proliferation is suppressed in low-oxygen conditions, and is exacerbated following the reduced oxygen supply. The cross-oxygen gradient comparisons suggest that MAPK and Hippo, which are core pathways of mammary cell proliferation, are repressed by hypoxia via oxidative-stress-dependent signals.

## 1. Introduction

Oxygen plays a vital role in regulating mammalian cells. Oxygen-availability-dependent alterations in cellular metabolism have profound functionality regulation effects in many tissues/organs, such as fetal development, dysfunction of the mucosal surface of the gastrointestinal tract, contractile activity maintenance of skeletal muscle, and blood vessel integrity. The adaptation or dysfunction of metabolic tissues in response to oxygen alteration results in metabolic pattern changes, subcellular structure switch, and cellular differentiation/proliferation [1]. The control of the cellular response to oxygen alteration involves a large group of specific genes, especially those regulated by *HIF* (hypoxia-inducible factor). HIF-1 is a crucial oxygen-sensing transcription factor regulator, consisting of HIF-1α and -1β subunits. In plentiful oxygen, HIF-1α is hydroxylated by prolyl hydroxylase domains (PHDs), and is then recognized and ubiquitinated by a ubiquitin ligase complex (pVHL), leading to its degradation via the proteasome. When the oxygen supply to cells is inadequate, HIF-1α is not hydroxylated due to decreased PHD activity, leading to HIF-1α accumulation and stabilization to form a functional transcription factor with HIF-1β. It then binds to the hypoxia response elements (HREs) in gene promoters, regulating the expression of the downstream genes [2]. Clarifying how cells respond to oxygen changes in the microenvironment is the premise for understanding the aforementioned biological process.

There is a growing number of studies concerning hypoxia in the mammary glands of mammals. Studies have shown that hypoxia in the mid-pregnant mammary gland can increase the protein expression of HIF-1 and promote GLUT1 expression in mice [3]. Increased oxygen consumption from fast-proliferating cells leads to hypoxia in breast cancer [4]. However, these studies were either concerned with pathological breast changes or potential implications for mammary development. Moreover, the milk synthesis and secretion process consume large amounts of oxygen. Some studies have found that hypoxia occurs during the lactation period in the mammary glands. Bovine mammary glands spontaneously develop chronic hypoxia to accommodate lactation and high metabolic load [5]. Mammary oxygen uptake is significantly higher in early lactation than in late pregnancy, increasing steadily over this period in dairy goats [6]. In addition, a positive correlation between milk yield and oxygen uptake has been identified in several observations [7,8]. A tissue biopsy study showed higher mRNA expression of HIF-1α during early and late lactation than other stages throughout the lactation period [9], suggesting a potential role of hypoxia in interplaying mammary metabolism. However, the effect of hypoxia on lactation in the mammary gland and its specific mechanisms have not yet been revealed. 

The mammary gland is a complex organ composed of several cell types: epithelial cells, adipocytes, perivascular cells, immune cells, and fibroblasts. Mammary epithelial cells (MECs), which proliferate and differentiate to form alveolus, are significant in fulfilling lactation functions [10]. Studies have demonstrated that milk secretion depends on the amount of MECs in the mammary gland, and that MEC proliferation is an essential determinant of mammalian lactation capacity [11]. The bovine primary MECs (bMECs) of dairy cows are an ideal cell model for investigating how hypoxia affects proliferation in milk-secreting cells. Moreover, for this study, it was necessary to set up different oxygen concentration gradients to mimic the fluctuating in vivo hypoxic microenvironment. Thus, the study’s objective was to clarify the effect of hypoxia and its underlying mechanisms on mammary cell proliferation under various oxygen concentrations. Outcomes from the current study can be used to enhance and maintain better lactation performance.

## 2. Materials and Methods

### 2.1. Isolation of Primary Bovine Mammary Epithelial Cells and Cell Culture 

Mammary tissues of Holstein cows were taken from an abattoir (Wuxi, China) which had already obtained permission for sampling. The bMECs were isolated as described below. The fresh tissues were rinsed with 0.3% bromo-geramine and D-Hank’s solution with 100 mg/mL gentamicin and 1000 IU/mL penicillin–streptomycin successively. The tissue was diced into small pieces. The cut tissues were digested with 0.25% trypsin at 37 °C for 30 min, and then the supernatant was removed. Subsequently, 5% collagenase solution, at final concentrations of 130 IU/mL type I and 138.5 IU/mL type II, was added, and digestion continued for 3–5 h at 37 °C. The cell suspension was filtered, and cells were harvested following centrifugation at 1000 rpm for 5 min. Cells were first resuspended in D-Hank’s solution, then centrifuged a second time, then resuspended in medium and cultured for 24 h in the presence of 5% CO_2_ at 37 °C. Cells were seeded below a density of 0.8 × 10^4^ cells per well in 96-well plates before hypoxic treatment. Finally, cells were cultured under various oxygen concentrations (1, 6, 11, 16, and 21% O_2_) for 24 h, and each treatment included five replicates.

### 2.2. Cell Proliferation Assay 

A colony formation assay was used to measure cell proliferation after 24 h of culture under different oxygen concentrations. After 24 h of cell culture, cells were inoculated in a 3.5 cm petri dish with 200 cells per dish. The cells were dispersed uniformly in a petri dish and incubated at 37 °C, 5% CO_2_, and saturated humidity for about 14 days until visible clones appeared. Then, cells were washed with phosphate buffered solution (PBS) twice and fixed with 4% paraformaldehyde (Shanghai Runjie Chemical Reagent Co., Ltd., Shanghai, China) for 15 min, and the fixing solution was removed. An appropriate amount of crystal violet solution (CAT No. C0003, Shanghai Biyuntian Biotechnology Co., Ltd., Shanghai, China) was added for 10 min, and then slowly washed away with running water. The petri dish was inverted to obtain the image and count the clones.

Cell viability was also assessed at 0, 6, 12, 18, and 24 h of cell culture under different oxygen levels by cell counting kit-8 (CCK8) (C0037, Biyuntian Biotechnology Co., Ltd., Shanghai, China). The cells in 96-well plates were added to the CCK-8 solution (20 mL), then incubated for an additional 3 h under the original culture conditions. The 450 nm optical densities of the plates were detected using a microplate reader.

### 2.3. The Adenosine Triphosphate (ATP) Level Measurement

The intracellular ATP levels were measured using the ATP Detection Kit (BC0305, Solarbio Science & Technology Co., Ltd, Beijing, China). The specific determination method followed the manufacturer’s instructions.

### 2.4. The Extracellular Oxygen Consumption Rates (OCR) Measurement

The extracellular OCR was detected using an Extracellular Oxygen Consumption Assay kit (ab197242, Abcam Inc., Cambridge, MA, USA). Cells to be tested were seeded in 96-well plates. The cells were washed with PBS, and then 150 μL of fresh medium was added. The 10 μL extracellular O_2_ consumption reagent was added and mixed thoroughly. Then, 100 μL of high-sensitivity mineral oil was added to each well to seal it, and the cell plate was used for fluorescence detection to analyze the real-time kinetics of OCR with the ratio EX (excitation wavelength)/EM (emission wavelength) = 380/650 nm. The rate of signal for each sample was calculated to obtain the OCR. 

### 2.5. Electron Microscopy

The original culture solution was discarded after 24 h, and the cells were fixed using paraformaldehyde for 2–4 h at 4 °C. The cells were pelleted by means of low-speed centrifugation, encapsulated with 1% agarose, and then washed thrice for 15 min with phosphate-buffered saline (PBS). Under different concentrations of alcohol and acetone, the cell samples were sequentially dehydrated. After being dehydrated, the cells were soaked in a mixture of Spurr resin and absolute acetone at ratios of 1:1 and 3:1 for 1 h and 3 h, respectively, and subsequently embedded with Spurr resin alone overnight. The embedded sample was heated to 70 °C overnight; then, 70–90 nm sections were obtained using an ultramicrotome (Leica, Wetzlar, Hesse, Germany, EM UC7). Lead citrate and uranyl acetate in 50% ethanol were used to stain the ultrathin sections for examination under a HITACHI H7650 transmissive electron microscope (TEM).

### 2.6. Mitochondrial Mass Analysis

MitoTracker Green (Biyuntian Biotechnology Co., Ltd., Shanghai, China) was diluted to a final concentration of 20~200 nM (1 mM stock dissolved in DMSO). The pre-warmed solution at 37 °C was added to the bMEC samples. After 15–45 min of co-incubation, the fresh culture solution was replaced at room temperature, and the fluorescence intensity of the cells was observed via fluorescence microscopy (OLYMPUS, Tokyo, Japan). The fluorescence intensity was calculated using ImageJ software version 2.0.0.

### 2.7. ELISA 

The superoxide dismutase (SOD; #A001-1-2), glutathione peroxidase (GSH-Px; #A005-1-2), malondialdehyde (MDA; #A003-1-2), total nitric oxide synthase (T-NOS; # A014-2-1), and total antioxidant capacity (T-AOC; #A015-3-1) were detected using enzyme-linked immunosorbent assay (ELISA) kits. The specific determination methods were conducted according to the manufacturer’s instructions (Nanjing Jiancheng Bioengineering Institute, Nanjing, China).

### 2.8. Flow Cytometric Analysis

The fluorescent probe Dichlorodihydrofluorescein diacetate assay (DCFH-DA) (Solarbio Science & Technology Co., Ltd, Beijing, China) was used to monitor the production of intracellular reactive oxygen species (ROS). ROS in cells oxidized fluorescent DCFH to produce fluorescent DCF, which could be detected. Cells with different oxygen levels were incubated in DCFH-DA (10 μM) (37 °C, 20 min) and washed in PBS. And the intensity of the DCF signal was detected to measure the ROS levels.

### 2.9. RNA Preparation, Illumina Sequencing, and RNA-Seq Analysis

Total RNA from frozen pbMECs was extracted under different oxygen levels to construct mRNA libraries for five groups, with five replicates per group. A total of 25 samples were sequenced via the Illumina HiSeqTM 2500 sequencing platform at Novogene Biotechnology Company (Tianjin, China). The primary sequencing data were subjected to quality control, and the cutadapt procedure included three steps, as shown below: (1) reads containing adaptors were removed; (2) reads less than 75 bp long after trim removal were removed; and (3) reads at the 5′ or 3′ ends lower than 20 bases, as well as the reads containing more than 5% of nitrogen, were removed. After filtering, clean data were aligned to the cow reference genome. The data were standardized with the fragments per kilobase of transcript per million mapped reads (FPKM) method, and the formula for the FPKM values was as follows: $FPKM=\frac{Total\;exon\;fragments}{Mapped\;reads\;(millions)\;\times \;exon\;length\;(Kb)}$. Furthermore, FPKM values were used in the gene expression analysis. The raw data were uploaded to the NCBI database.

Principal component analysis (PCA) was implemented in R version 4.2.0 using the FactoMineR and factoextra packages. Differentially expressed genes (DEGs) were calculated and analyzed using the R package edgeR. A volcano plot of DEGs was rendered using the ggthemes and ggpubr R packages, with a false discovery rate (FDR) < 0.05 and |log2fold change| > 1 as a filter criterion. A bioinformatics and evolutionary genomics web tool was used to generate a Venn diagram. A soft-clustering analysis was performed with the R package Mfuzz based on the fuzzy c-means algorithm. The converted Entrez Gene IDs were obtained, then assigned to the Kyoto Encyclopedia of Genes and Genomes (KEGG) pathways using the package org.Bt.eg.db. The bubble plots of the top 20 KEGG enrichment pathways were drawn with the ggplot2 package. Heatmaps of DEGs from the key signaling pathways were generated with the R package pheatmap. The protein–protein interaction (PPI) analyses were carried out using the Search Tool for the Retrieval of Interacting Genes/Proteins (STRING) web (version 11.5; https://string-db.org/, accessed on 5 February 2023), and the networks were generated using Cytoscape software (version 3.9.1) to screen the key genes.

### 2.10. Real-Time Quantitative Polymerase Chain Reaction (qPCR)

The mRNA expression levels of five core genes, NGFR, CSF1, KDR, IL1R1, and PPP2R2B, from RNA-seq analysis were detected using the qPCR method. Total RNA was extracted using Trizol lysate, with reference to the TIANGEN Reverse Transcription kit (TIANGEN Biotech Co. Ltd., Beijing, China). The RNA concentration and purity were measured using a NanoDrop2000 spectrophotometer (NanoDrop2000, Thermo Fisher Scientific, Waltham, MA, USA), and the ratios of A260/A280 and A260/A230 were used for evaluation. Subsequently, qPCR was conducted using AceQ qPCR SYBR Green Master Mix (Vayzme, Nanjing, China). The primer sequences are provided in Appendix A. Relative quantification of mRNA levels was based on the reference gene β-actin. Quantitative results of mRNA expression were calculated using the delta-delta Ct method (2^−ΔΔCt^) [12]. Five biological replicates and three technical replicates were detected.

### 2.11. Western Blot Analysis 

HIF-1α, dynamin-related protein 1 (DRP1), and optin atrophy protein 1 (OPA1) expression were detected to analyze the hypoxic response and mitochondrial mass. The phosphorylation of ERK and JNK from the MAPK signaling pathway were determined. The bMECs were lysed by radioimmunoprecipitation assay (RIPA) lysates, and then the supernates were harvested by centrifugation at 12,000 rpm for 10 min. According to the instruction, a bicinchoninic acid (BCA) kit (Beyotime Institute of Biotechnology, Shanghai, China) was utilized to access the protein concentration. The supernatant was mixed with the appropriate amount of 5X sodium dodecyl sulfate (SDS) loading buffer, then heated in a boiling water bath for 5 min. After preparing 10% separation adhesive and 5% concentration adhesive, the sample was added to each well in an electrophoresis tank for sample loading and then transferred onto polyvinylidene fluoride (PVDF) membranes. After washing three times (5 min/wash) with Tris-buffered saline and Tween 20 (TBST), incubation of the membranes was performed with the primary antibody overnight at 4 °C. After the incubation, corresponding secondary antibodies were then added and incubated for 1 h, followed by three additional rinses three times at 5 min intervals. Finally, target protein expression was detected using electrochemiluminescence (ECL) luminous fluid, and bands were visualized using Quantity One software (version 4.6.2). Blot bands were analyzed using ImageJ software. The following primary antibodies were used: anti-HIF-1α (#AF1009, Affinity Biosciences, Cincinnati, OH, USA), anti-DRP1 (#HA500487, HUABIO Co., Ltd, Hangzhou, China), anti-OPA1 (#ET1705-9, HUABIO Co., Ltd, Hangzhou, China), anti-ERK1/2 (#4695S, Cell Signaling Technology [CST], Danvers, MA, USA), anti-phospho-ERK1/2 (#9101s, Cell Signaling Technology [CST], Danvers, MA, USA), anti-JNK1/2 (#9252s, Cell Signaling Technology [CST], Danvers, MA, USA), anti-phospho-JNK1/2/3 (#AF3318, Affinity Biosciences, Cincinnati, OH, USA), and anti-GAPDH (#60004-1-Ig, Proteintech, Inc., Wuhan, China). The secondary antibodies (#A0208 and #A0216) were purchased from Shanghai Biyuntian Biotechnology Co., Ltd., Shanghai, China.

### 2.12. Statistical Analysis

Statistical analyses were performed using GraphPad Prism version 8.4.0. The unpaired *t* test and one-way analysis of variance (one-way ANOVA) test compared two and multiple groups, respectively. Origin Pro 2017 software (OriginLab, Northampton, MA, USA) was used for the linear regression analyses and curve plotting. *p* < 0.05 was considered a statistically significant difference, and *p* < 0.01 indicated an extremely significant difference.

## 3. Results

### 3.1. Proliferation Characters, Oxygen Utilization, and Oxidative Stress of bMECs with Different Oxygen Availability 

Cell proliferation was measured by means of clonal formation. The cell cloning formation results of bMECs cultured under five oxygen concentration gradients are shown in Figure 1A. Compared with the 21% group, the clone formation in the hypoxic group was significantly lower (*p* < 0.01), and the decrease was greater with the reduced oxygen concentration. The cell viability for further analysis of cell proliferation by CCK8 assay at 0, 6, 12, 18, and 24 h under different oxygen levels is presented in Figure 1B. Time-course analysis specified that the cell proliferation rate was altered by oxygen availability initiated 6 h post-treatment, and a higher degree of proliferation inhibition was observed following the treatment. Moreover, the reduced cell proliferation rate was accompanied by reduced oxygen availability from 21 to 1% (*p* < 0.01, Figure 1B). Indicators of oxygen utilization, including HIF-1α, oxygen consumption rate, cellular ATP, and mitochondrial function parameters, are shown in Figure 2. A linearly increased HIF-1α expression was detected, with gradually reducing environmental oxygen availability (*p* < 0.01, Figure 2A). Moreover, compared with 21% oxygen, the oxygen consumption rate (*p* < 0.01, Figure 2B) and ATP level (*p* < 0.01, Figure 2C) decreased linearly with the decline in oxygen concentration. The protein expression analysis showed higher DRP1 (*p* < 0.01) expression and lower OPA1 expression (*p* < 0.01) with a decrease in oxygen concentration (Figure 2D,E). A fluorescence microscopy study revealed that blue fluorescence corresponded to the staining of the nucleus, and the green fluorescent signals referred to the mitochondria (Figure 2F). Quantification of the fluorescence intensity displayed stronger fluorescence in the 1% oxygen concentration cells than in other groups (Figure 2G). TEM was used to observe the mitochondrial structure and to detect morphological changes. Mitophagy appeared under 1% oxygen concentration, but the other four groups did not exhibit this phenomenon (Figure 2H). The levels of ROS, MDA, and the antioxidant enzymes are presented in Figure 3. Linear increases in ROS (*p* < 0.01) and MDA (*p* < 0.01, Figure 3A,B) were noted when the oxygen concentration progressively decreased. In contrast, the activities of SOD (*p* < 0.01), GSH-PX (*p* < 0.01), T-AOC (*p* < 0.01), and T-NOS (*p* < 0.01) were reduced linearly with the decreasing oxygen concentration, and remained consistently positive (Figure 3C–F). 

### 3.2. Transcriptome Analysis

Transcriptomic analysis of the mRNA expression profile of bMECs under five oxygen levels, with five biological replicates, was performed to investigate how different levels of oxygen availability would impact bMEC proliferation. Following the raw reads filter steps, the averages of 4.4 × 10^7^ clean reads were obtained. After the quality assessment, the average guanine–cytosine (GC) content and Q30 ratio were 53.93 and 94.17%, respectively (Appendix A). All the raw data were submitted to the NCBI database (BioProject: PRJNA935409). Furthermore, PCA was performed to visualize the comprehensive gene expression pattern among the five groups of bMECs (Figure 4A). The results demonstrated that the between-class PCA separated the five groups, and the 1% group was clearly segregated from the other groups. We performed separate pairwise comparisons between adjacent treatment groups in order to investigate the DEGs across cells treated with gradually decreasing oxygen concentrations. Different numbers of DEGs were detected across different treatments: 505 between 1% and 6% (Figure 4B), 739 between 6% and 11% (Figure 4C), 60 between 11% and 16% (Figure 4D), and 388 between 16% and 21% (Figure 4E), respectively. There were 1256 DEGs across the four pairwise comparisons of five groups (Figure 5A). Moreover, to examine the dynamic expressional alteration of DEGs under different oxygen degrees, we performed clustering analysis to reflect similar expression patterns using the Mfuzz package. The results found that 1256 DEGs were sorted into 12 clusters, including 2 down-regulated clusters (clusters 1 and 7) and 10 other mixed clusters (Figure 5B). The KEGG analysis was performed for each cluster using the KEGG database, and the key signaling pathways were selected. They are listed according to their functions in Table 1 and Appendix A, including 4, 5, 8, 9, and 10 clusters, respectively (*p* < 0.05). MAPK and Hippo signaling pathways, which are two common pathways concerned with cell proliferation, were enriched in clusters 4 and 9, respectively (*p* < 0.05). The following are the other pathways associated with oxidative stress that are significantly activated by different degrees of hypoxic conditions: the p53 (*p* < 0.001), HIF-1 (*p* < 0.001), and PI3K-Akt signaling pathways (*p* < 0.05), as well as phagosome (*p* < 0.001), enriched in clusters 5, 8, 9, and 10, respectively. Their FPKM values are listed in Appendix A, respectively. MAPK and Hippo signaling pathways were strongly linked with cell proliferation among these oxygen-sensitive transcripts. The heatmap displayed the dynamic expression alteration of genes in the representative pathways in response to oxygen alteration (Figure 6A). The results illustrated that gene expression was suppressed from the MAPK and Hippo signaling pathways in four hypoxia groups compared with cells treated with 21% oxygen. Regarding dynamic changes in signaling pathways in response to oxygen alteration, the HIF-1 sand PI3K-Akt signaling pathways had the highest expression levels at 1% and 6%, respectively. The genes from the p53 signaling pathway and phagosome were most highly expressed at 11% and 16%. The higher expression level of DEGs indicated more activation under the corresponding oxygen levels. 

Next, we explored the interplay between oxidative stress and cell proliferation pathways by importing the PPI network analysis of DEGs-significant pathways into the STRING database. We also analyzed the PPIs, which were highly expressed corresponding to a certain oxygen concentration (Figure 6B–D). The direct relation between genes was evident in the PPI network. Gene–gene interaction analysis revealed an interaction between the HIF-1 and MAPK signaling pathways under the lowest oxygen concentration (1% oxygen). KDR, CSF1, and IL1R1 were the core genes in the inhibited MAPK signaling pathway, and the GAPDH and SERPINE1 genes from the pathway were highly expressed and interacted with KDR, CSF1, and IL1R1 (Figure 6B). The PPI network analysis exposed an interaction among genes between the PI3K-Akt and MAPK signaling pathways at an oxygen concentration of 6%. Two core genes, KDR and CSF1, of the MAPK signaling pathway were inhibited, which interacted with the PDGFD, PDGFRA, and COL1A2 genes from the activated PI3K-Akt signaling pathway in the network (Figure 6C). Additionally, the p53 signaling pathway genetically interacted with the Hippo signaling pathway under 11% and 16% oxygen concentrations. PPP2R2B, the key gene in the Hippo signaling pathway, was inhibited and interacted with the CCNB2, CCNB1, and CDK1 genes in the activated pathway (Figure 6D). Core genes in the MAPK and Hippo pathways from RNA-seq interaction networks were further validated for mRNA expression levels. As shown in Figure 7, NGFR, KDR, and PPP2R2B in the 1% to 16% group were significantly lower than in the 21% O_2_ (*p* < 0.01). Compared with the normal oxygen group, CSF1 was significantly reduced at a 6% oxygen concentration (*p* < 0.05), and was significantly decreased in the 1%, 11%, and 16% groups (*p* < 0.01). The IL1R1 was significantly lower in the 1%, 11%, and 16% groups than in the nor-oxygen group (*p* < 0.05). In order to verify that the MAPK signaling pathway was indeed inhibited under hypoxic conditions compared to 21% O_2_, the phosphorylation levels of the core genes REK and JNK, which played important regulatory roles in the pathway, were detected (Figure 8). The results suggested that phosphorylation levels of ERK1/2 and JNK1/2/3 were inhibited at 16% to 1% hypoxia compared to 21% O_2_.

## 4. Discussion

The proliferation of MECs plays a vital role in maintaining lactation persistency in dairy cows [13]. Although previous research has suggested that oxygen-responsive genes in the mammary glands of cows at different stages of lactation are altered [9], the role of oxygen in lactation persistency has not been defined. For the first time, we investigated the proliferation traits of bMECs at various degrees of oxygen availability. By learning about the proliferation traits, oxygen utilization, oxidative stress index, and transcriptomic profiles and validation of relevant key genes, we provided the key proliferative-regulatory molecules responding to oxygen deficiency. The outcomes of the present study may contribute to regulating lactation persistency via mediating mammary proliferation.

A previous study revealed that stability and accumulation of HIF-1α protein increased in an oxygen-concentration-dependent manner in cells [14]. Some studies have also demonstrated pathologic hypoxia-induced proliferation inhibition of various types of cells, including hematopoietic stem cells, embryonic fibroblasts, and lymphocytes [15,16]. In the current study, when the oxygen availability decreased from 21% to 1%, the protein expression of HIF-1α increased gradually, suggesting a vital role of HIF-1α in responding to mammary oxygen deficiency in the mammary gland. Upon the appearance of environmental hypoxia, reduced aerobic respiration and the enhancement of anaerobic respiration can lead to a decrease in cellular ATP generation and oxygen consumption [2], which is consistent with our study. In addition, HIF-1α can also inhibit the activity of key enzymes in cellular energy metabolism by directly regulating the expression of HIF-dependent genes, which further leads to cell energy deficiency [17]. Following the gradually altered expression of HIF-1α, we observed a time- and dose-dependent proliferation inhibition due to oxygen deficiency, suggesting the occurrence of proliferation inhibition by hypoxia via different metabolic routes, such as inflammatory cytokine secretion and oxidative damage induction [18]. This finding indicates that the hypoxic environment was adverse to the periodic renewal of mammary epithelial cells, which can have a potential negative impact on the maintenance of lactation function. Furthermore, we evaluated redox status alteration under different levels of oxygen availability. The gradual increase in ROS and MDA and the reduction in antioxidant enzymes (SOD, GSH-Px, T-AOC, and T-NOS) following gradient hypoxia were associated with cellular proliferation activity. Many studies have stated that HIF-1α and ROS from mitochondrial sources regulate each other positively. The HIF-1α can induce ROS generation, predominantly depending upon several oxygen-dependent enzymes, including nicotinamide adenine dinucleotide phosphate (NADPH) oxidase, cytochrome c oxidase, and uncoupled endothelial nitric oxide synthase (eNOS) [19,20], and ROS production inactivates PHD2 and stabilizes HIF-1α [21]. The oxidative stress was also associated with imbalanced mitochondrial fusion–fission, modulating the gene expression of DRP1 and OPA1 [22] and causing mitophagy [23]. This is consistent with our study in that the excessive generation of ROS could result in a decline in cell function and cause progressive cell damage [24,25]. Moreover, oxidative stress was connected with inhibiting cell proliferation, and hydrogen peroxide (H_2_O_2_) treatment was able to suppress cell proliferation [26,27]. This evidence suggests the occurrence of hypoxia-induced proliferation inhibition of the mammary gland via impairing its antioxidant system.

The DEGs of the MAPK and Hippo signaling pathways were repressed under 16% oxygen concentrations and lower, compared with 21%, which was the primary cause for the cell proliferation phenotype repression under the four hypoxia group. We found that decreased PPP2R2B of the Hippo signaling pathway was highlighted in the PPI network and associated with the other genes from the p53 signaling pathway under 16% and 11%. A previous study demonstrated that PPP2R2B regulates the nuclear localization and stability of YAP/TAZ [28], two core components of the kinase cascade, accumulating and entering the nucleus to regulate the gene expressions involved in cell proliferation [28,29]. The post-translational modifications of the p53 protein were up-regulated under obvious levels of oxidative stress [30], and activated p53 signaling further promoted the activity of ROS-generating enzymes, inhibiting the promoter expression of manganese superoxide dismutase (MnSOD) [31,32]. Thus, the activated p53 signaling pathway was pivotal in coordinating oxidative stress to suppress cell proliferation through genetic interactions with PPP2R2B from the Hippo signaling pathway at 16% and 11% oxygen. The interaction between the PI3K-Akt and HIF-1 signaling pathways and the MAPK signaling pathway at 6% and 1% concentrations suggested that the altered oxygen availability changed the cellular proliferation through a tertiary kinase cascade-mediated mechanism [33]. The three-tiered kinase cascade has already been well established; that is, the signaling molecules preferentially activate Ras and then phosphorylate MAP3K (Raf), which subsequently activates MAP2K (MEK) and eventually activates MAPKs (Erk1/2) [34,35]. We confirmed the molecular changes in these pathways through RNA-seq and q-PCR approaches, including KDR, CSF1, and NGFR from the MAPK signaling pathway, which is consistent with their previous reported function in cellular proliferation [33]. The ERK1/2 and JNK1/2/3 subfamilies are crucial components of MAPK cascade, and their decline in phosphorylation further clarified the inhibition of the MAPK pathway under hypoxia. According to the findings of this study, genes from the PI3K-Akt signaling pathway were up-regulated under 6%, and the pathway has been shown to be allied with oxidative stress; that is, ROS can directly activate PI3K in cells [36,37]. Moreover, in the gene network generated under a 6% oxygen level, KDR and CSF1 were low-expressed molecules playing a key role in the MAPK signaling pathway, interacting with HIF-1 signaling under 1% and resulting in the oxidative stress of mammary gland and declined mitochondrial function [21]. Accordingly, the results of the current analysis clarify that cell proliferation was inhibited under four different levels of hypoxia due to the inhibition of the core pathways involved in cell proliferation, i.e., the Hippo and MAPK signaling pathways, which were primarily regulated by activated pathways and genes related to oxidative stress from oxygen concentrations of 16–1%.

## 5. Conclusions

This current research investigated the proliferation changes in bMECs and the underlying mechanism through time- and dose-scaled oxygen deficiency. We found that the bMECs developed an imbalance in their oxidant–antioxidant status that subsequently led to oxidative stress, which was accompanied by the decline in mitochondrial mass and function in response to a decrease in available oxygen. In addition, a linearly reduced bMEC proliferation rate occurred following the gradient decrease in oxygen concentration. By conducting transcriptomes of MECs and q-PCR validation, we found that two key proliferative pathways, including the MAPK and Hippo signaling pathways, were the canonical pathways responsible for the gradient oxygen deficiency. Furthermore, the inhibition of these two signaling pathways, MAPK and Hippo, can be mainly attributed to their key regulators, including PPP2R2B, KDR, CSF1, and NGFR. Finally, the interplay of these key regulators with oxidative-stress-related pathways led to the MAPK- and Hippo-associated proliferation inhibition of bMECs. The current study’s outcomes may help in the development of an oxygen metabolic-based approach to improve the lactation persistency of dairy cows, thereby increasing milk yield and milk quality during lactation.

## Figures and Tables

**Figure 1 antioxidants-13-00288-f001:**
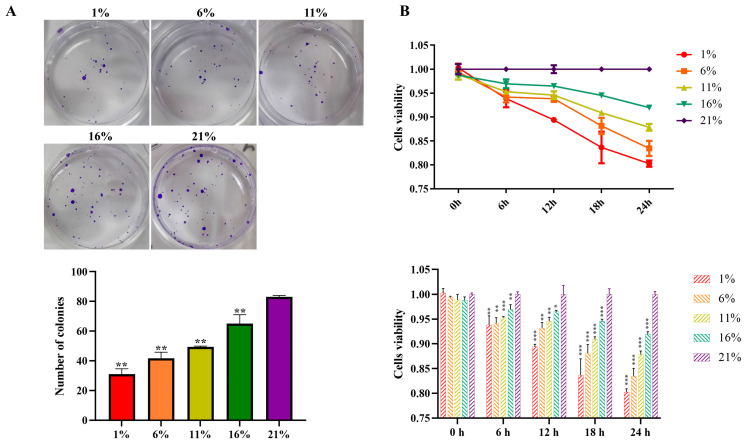
Cell proliferation under different oxygen concentrations. (**A**) Clone formation. The upper part displays the images of clone formation. The lower part presents the quantitative results of the number of colonies, and all comparisons were made relative to the 21% group. (**B**) CCK8 assay. The upper part shows the change curve of cell viability with time and concentrations. The lower part is the analysis of differences between four hypoxic groups and the 21% group at each time point, respectively. The values of cell viability were calculated relative to the 21% group. * *p* < 0.05, ** *p* < 0.01, *** *p* < 0.001. Each bar represents mean ± SD.

**Figure 2 antioxidants-13-00288-f002:**
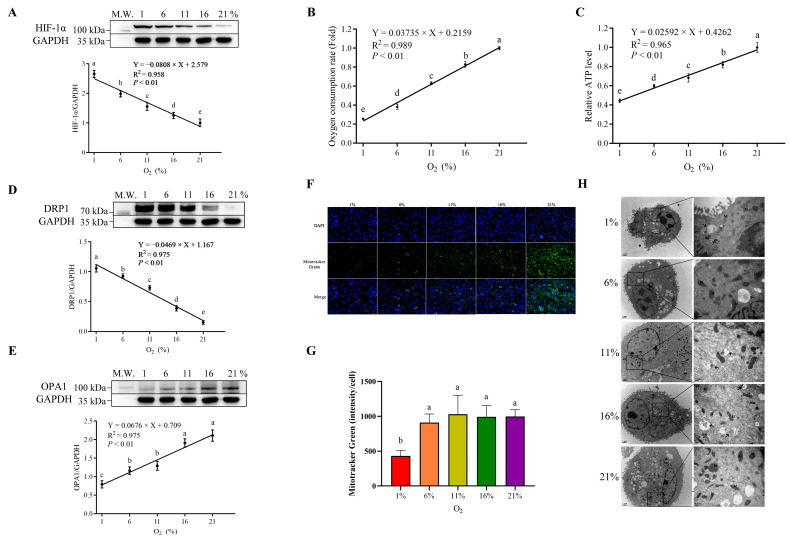
The relative indicators of oxygen utilization under different oxygen concentrations. (**A**) Protein expression of HIF-1α under different oxygen concentrations was measured; the upper panel shows Western blotting banding pictures, and the lower panel shows their quantification. (**B**) Oxygen consumption rate relative to the 21% group. (**C**) The cellular relative ATP level relative to the 21% group. The mitochondrial function (**D**–**H**) was assessed. Western blot analysis of DRP1 (**D**) and OPA1 (**E**) was determined. Cells were stained with Mito-tracker Green to visualize the mitochondria and Hoechst 33,342 to visualize the nucleus. (**F**,**G**) The higher relative fluorescence intensity represents an increase in mitochondrial mass. (**H**) The TEM of mitochondrial morphology under different oxygen levels. M, mitochondria; N, nucleus; ER, endoplasmic reticulum. Single arrow, phagosome; double arrow, mitophagy. Different letters (a–e) indicate significant differences among treatments based on a *p* value < 0.05. Each bar represents mean ± SD.

**Figure 3 antioxidants-13-00288-f003:**
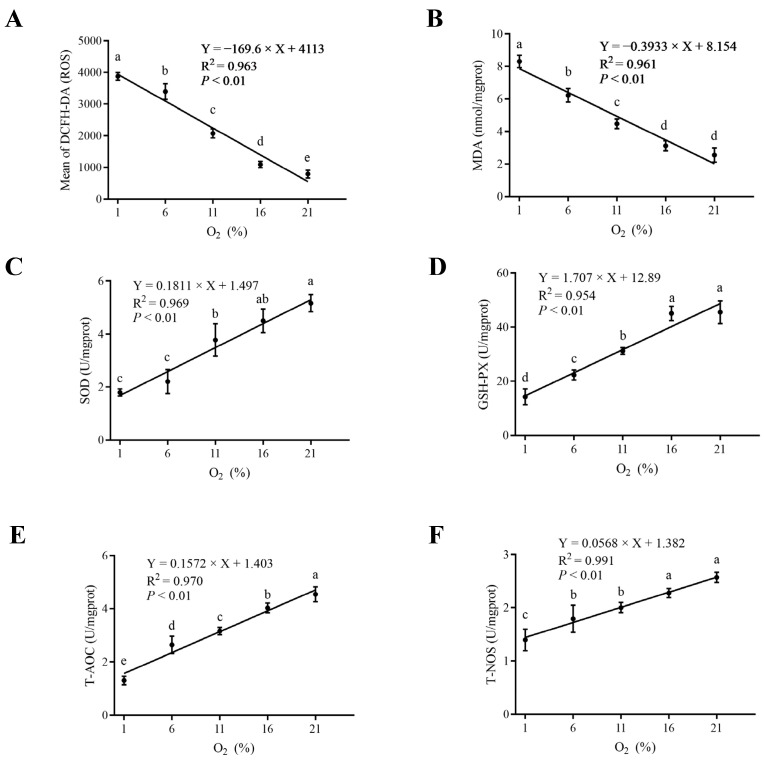
Oxidative stress indices under different oxygen concentrations. The levels of ROS (**A**), MDA (**B**), SOD (**C**), GSH-PX (**D**), T-AOC (**E**), and T-NOS (**F**). Different letters (a–e) indicate significant differences among treatments based on a *p* value < 0.05. Each bar represents mean ± SD.

**Figure 4 antioxidants-13-00288-f004:**
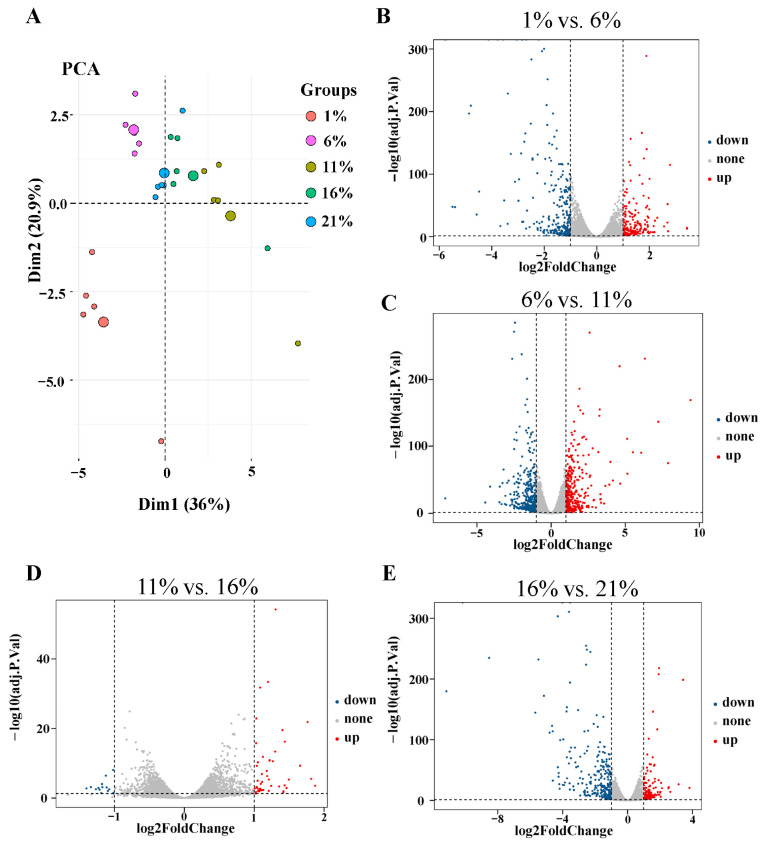
Differentially expressed genes in bMECs under different oxygen levels. (**A**) Principal component analysis (PCA) of five groups based on all detected genes in RNA-seq. Volcano plot for mRNAs in (**B**) 1% vs. 6%, (**C**) 6% vs. 11%, (**D**) 11% vs. 16%, and (**E**) 16% vs. 21%. The vertical dotted line delimits up-(red) and down-(blue) regulation (|log2-fold change| > 1, FDR < 0.05).

**Figure 5 antioxidants-13-00288-f005:**
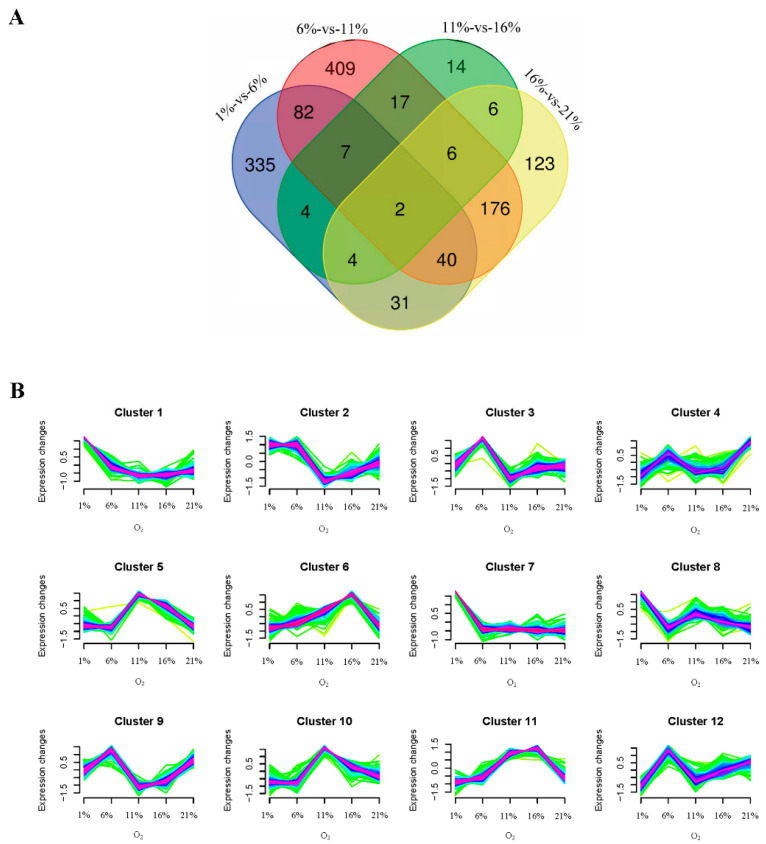
Cluster analysis of differential gene expression. (**A**) Unique and shared differential expression genes in 1–6%, 6–11%, 11–16%, and 16–21% pairwise analysis. (**B**) Soft clustering analysis from differential genes of all the comparison groups, where each line represents the expression change of one gene. Purple/blue color indicates stronger membership, and green/yellow color denotes weaker membership.

**Figure 6 antioxidants-13-00288-f006:**
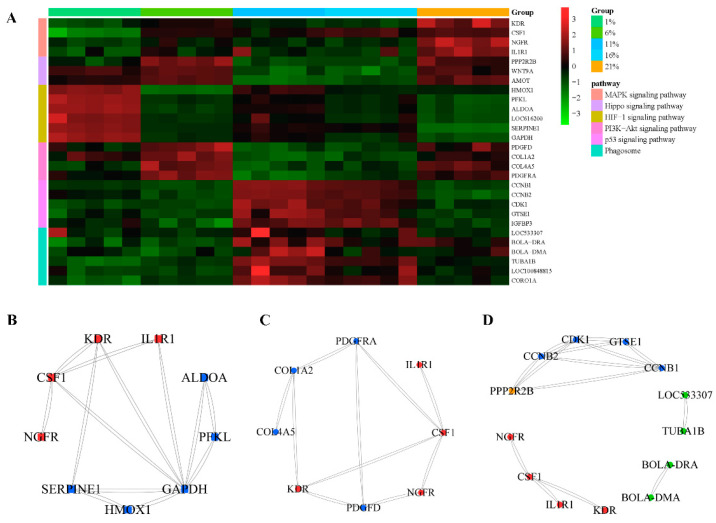
Analysis of key gene networks under different oxygen levels. (**A**) The gene expression heatmap of the key pathways. Blue indicates a decrease and red indicates an increase in gene expression (see color set scale on top right corner). In protein–protein interaction (PPI) network diagrams, the color of the dots represents: (**B**) 1% O_2_, red (MAPK signaling pathway), and blue (HIF-1 signaling pathway); (**C**) 6% O_2_, red (MAPK signaling pathway), and blue (PI3K-Akt signaling pathway); and (**D**) 11% and 16% O_2_, red (MAPK signaling pathway), orange (Hippo signaling pathway), blue (p53 signaling pathway), and green (phagosome).

**Figure 7 antioxidants-13-00288-f007:**
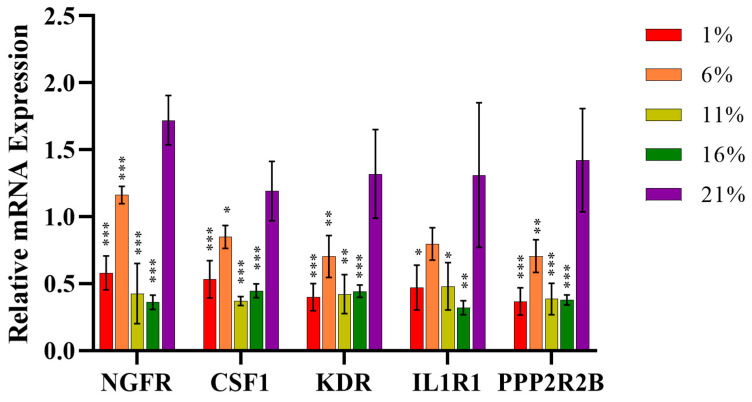
The RT-qPCR validation of the relative mRNA expression of core genes in MAPK and Hippo pathways. Among them, NGFR, CSF1, KDR, and IL1R1 were in the MAPK signaling pathway, and PPP2R2B was from the Hippo signaling pathway. ß-Actin was used as the reference gene. The comparisons were made relative to the 21% group. * *p* < 0.05, ** *p* < 0.01, *** *p* < 0.001. Each bar represents mean ± SD.

**Figure 8 antioxidants-13-00288-f008:**
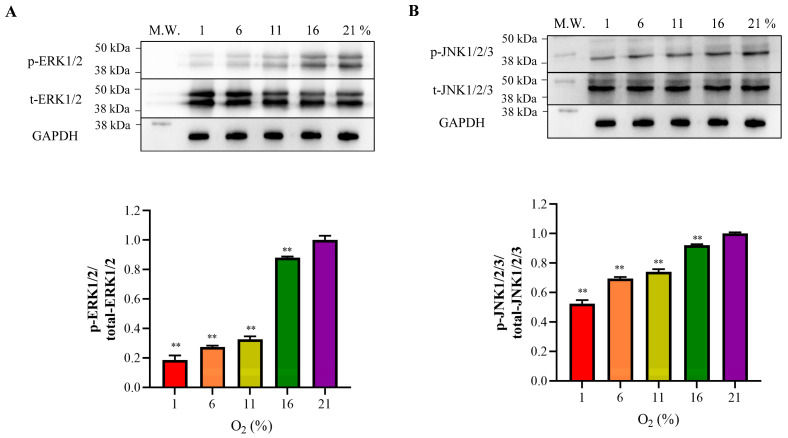
The phosphorylation of ERK1/2 and JNK1/2/3 as the key component of the MAPK family under different oxygen concentrations. (**A**) ERK activity was calculated from the ratio of phosphorylated ERK/total ERK relative to the 21% group. (**B**) JNK activity was calculated from the ratio of phosphorylated JNK/total JNK relative to the 21% group. The upper panel shows Western blotting banding pictures, and the lower panel shows their relative quantification. All comparisons were made relative to the 21% group. ** *p* < 0.01. Each bar represents mean ± SD. tERK: total ERK; pERK: phospho-ERK; tJNK: total JNK; pJNK: phospho-JNK.

**Table 1 antioxidants-13-00288-t001:** KEGG pathways involved in cell proliferation from clusters.

Cluster	KEGG ID	Pathway	*p* Value	*p* adj	Count
cluster4	bta04010	MAPK signaling pathway	0.030755	0.228955	4
cluster5	bta04115	p53 signaling pathway	0.000109	0.002291	5
cluster8	bta04066	HIF-1 signaling pathway	2.01 × 10^−6^	5.99 × 10^−5^	6
cluster9	bta04390	Hippo signaling pathway	0.049901	0.482730	3
	bta04151	PI3K-Akt signaling pathway	0.049885	0.482730	5
cluster10	bta04145	Phagosome	0.000434	0.006652	6

## Data Availability

The original contributions presented in this study are included in the article/Appendix A. Further inquiries can be directed to the corresponding author.

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
