# Peer review of "Cross-Oxygen Gradients Transcriptomic Comparison Revealed the Central Role of MAPK and Hippo in Hypoxia-Mediated Mammary Proliferation Inhibition"

_antioxidants, 2024, doi:10.3390/antiox13030288_

Round 1

Reviewer 1 Report (Previous Reviewer 2)

This is a new version of a previous manuscript. New experiments and determinations have been included to support the main hypothesis.

Although some determinations (clone assay is not a direct measurement of cell proliferation for example) could be complemented, there is enough data to support the conclusions.

The manuscript is fine.

Author Response

Reviewer 2 Report (Previous Reviewer 1)

There is not Major Comments

The reviewed version of this paper is acceptable for publication. However, there are some minor comments that need to be taken into account:

Section 2.2: Lines 93-95 authors use past continuous tenses and in lines 97-102 authors use infinitive “Wash” “Remove” “Invert”.

Section 2.4. Lines 112-114 authors use past continuous tenses and in lines 115 “Add”

Please write both sections properly.

Section 2.4. line 15, please O2 using subscript to write 2.

Author Response

Reviewer 3 Report (New Reviewer)

The study's objective was to clarify the effect of hypoxia and its underlying mechanisms on mammary cell proliferation under various oxygen concentrations; outcomes from that study could be used to enhance and maintain better lactation performance.

The manuscript describes important physiological mechanisms, which can be used to clarify pathways of milk synthesis and production. As such, it is very useful and merits publication.

Overall. The manuscript can be published after minor revision and appropriate improvement.

I have added some points below that can be addressed to improve the final version.

-The methodology section includes many technicalities which can be avoided to save the reader to go through these minute details. The authors can refer to previously published papers. Removing all these details will benefit the manuscript.

-Visualisation of results should be improved. For example, figures 1, 2, 3, 7, 8 can be colourised.

-The Discussion does not fully cover the whole topic of the study. Moreover, some recent relevant references, which will help the authors to explain better their findings, are missing. The authors can search and include them in the revise version.

-A final paragraph about the clinical significance of the results should be included.

Author Response

This manuscript is a resubmission of an earlier submission. The following is a list of the peer review reports and author responses from that submission.

Round 1

Reviewer 1 Report

Comments and Suggestions for Authors

The authors have presented their research work assessing the effect of Oxygen on cultured MECs. Results showed that cell proliferation decreases and hypoxic inducible factor increases. It is an excellent paper and it should be published after minor revisions that will contribute to a better understanding of the work.

11.  Some format details: Please revise subscript in text (i.e. lines 90 and 92 CO2 or O2)

22. Figure 1B is not clear; data in Table will be more useful to understand those values.

33. Related to Figure 1B, please explain in the text reasons for significant differences for t=0h between 1%, 11% and 16%, and those no differences between 1 and 21%.

44.  Please explain in text the effect of these differences in t=0 on the other studies carried out in this research work.

55. Please improve quality of Figure 4.

66.  Conclusions need to be improved including those findings obtained from the studies carried out.

Comments on the Quality of English Language

Good quality of English

Reviewer 2 Report

Comments and Suggestions for Authors

This is an interesting article, however, there are several limitations in the study that needs to be addressed before it is ready for publication. Some of the limitations are important and therefore, they need to be addressed with new results that support the conclusions.

Main points:

The so-called proliferation assay is a viability assay based on mitochondrial dehydrogenase activity. It is not a proliferation assay and therefore, a real proliferation assay should be presented. Furthermore, it should be taken into account that the authors indicate in the manuscript that a decrease in the number and functionality of the mitochondria have been detected, and therefore this fact prevents the use of this assay to address cell viability and proliferation.

The value of the TEM mitochondria images is limited. Assays, for example of ATP levels in the cells or isolated mitochondria respiration, should be included.

In addition, the involvement of MAPK, Akt pathways is postulated. The phosphorylative state of key kinases (by Western blot using phosphoantibodies) should be presented.

Minor points:

In Figure 1 values are relative compared with 21% oxygen. This should be indicated in the legend.

In de abstract, the word degree is misleading.

Comments on the Quality of English Language

Please check minor points in the manuscript.

Round 2

Reviewer 2 Report

Comments and Suggestions for Authors

Unfortunately, this new version has not addressed the problem of the previous one. In fact, no further data is presented, and furthermore, the main conclusion of the ms is based on the decreased mitochondrial activity that has been used to measure cell viability. Another main drawback that remains in place is the lack of information on the regulation of the signaling pathways involved since phosphorylative status is missing.

Comments on the Quality of English Language

OK